# Neurometals in the Pathogenesis of Prion Diseases

**DOI:** 10.3390/ijms22031267

**Published:** 2021-01-28

**Authors:** Masahiro Kawahara, Midori Kato-Negishi, Ken-ichiro Tanaka

**Affiliations:** Department of Bio-Analytical Chemistry, Research Institute of Pharmaceutical Sciences, Faculty of Pharmacy, Musashino University, Tokyo 202-8585, Japan; mnegishi@musashino-u.ac.jp (M.K.-N.); k-tana@musashino-u.ac.jp (K.-i.T.)

**Keywords:** synapse, amyloid, calcium homeostasis, neurotoxicity, Alzheimer’s disease, dementia with Lewy bodies

## Abstract

Prion diseases are progressive and transmissive neurodegenerative diseases. The conformational conversion of normal cellular prion protein (PrP^C^) into abnormal pathogenic prion protein (PrP^Sc^) is critical for its infection and pathogenesis. PrP^C^ possesses the ability to bind to various neurometals, including copper, zinc, iron, and manganese. Moreover, increasing evidence suggests that PrP^C^ plays essential roles in the maintenance of homeostasis of these neurometals in the synapse. In addition, trace metals are critical determinants of the conformational change and toxicity of PrP^C^. Here, we review our studies and other new findings that inform the current understanding of the links between trace elements and physiological functions of PrP^C^ and the neurotoxicity of PrP^Sc^.

## 1. Introduction

Prion diseases are fatal neurodegenerative diseases, including scrapie in sheep, bovine spongiform encephalopathy in cattle, chronic wasting disease in elk, and Creutzfeldt–Jakob disease (CJD), Gerstmann–Sträussler–Scheinker syndrome, and Kuru in humans. The pathological hallmarks of prion diseases are the spongiform degeneration of glial cells and neurons, synaptic degeneration, and the accumulation of abnormal scrapie-like prion protein (PrP^Sc^) in the brain [1]. It is widely accepted that the conformational conversion of normal cellular prion protein (PrP^C^) to pathogenic PrP^Sc^ is central in the pathogenesis of these diseases. Although PrP^C^ and PrP^Sc^ have the same chemical characteristics and primary sequence, PrP^Sc^ differs from PrP^C^ in terms of its high content of β-sheet secondary structure, propensity to form insoluble amyloid fibrils, and resistance to protease digestion. When misfolded PrP^Sc^ enters the body via the ingestion of contaminated food or iatrogenic contamination, the protease-resistant PrP^Sc^ invades the brain, forms aggregates and amyloid fibrils, and in turn promotes neighboring PrP^C^ molecules to misfold and aggregate. Thus, prion diseases are also called transmissible spongiform encephalopathy (TSE). 

In this sense, prion diseases are included in the category of conformational diseases (protein-misfolding diseases), including Alzheimer’s disease (AD) and dementia with Lewy bodies (DLB) [2]. All of these diseases share common properties, such as the deposition of disease-related proteins and exhibition of neurotoxicity. These disease-related proteins, termed “amyloidogenic proteins”, include β-amyloid protein (AβP) in AD, PrP in prion diseases, and α-synuclein in DLB. Although their primary sequences are different, these proteins commonly form insoluble fibril-like structures (amyloid fibrils) with β-pleated sheet structures. AβP is a small peptide consisting of 39–43 amino acid residues that is secreted by the cleavage of a large precursor protein (APP; amyloid precursor protein). The conformational change of AβP and its neurotoxicity play central roles in the pathogenesis of AD [3]. The accumulation of Lewy bodies is observed in patients with DLB and other diseases such as Parkinson’s disease (PD) and multiple system atrophy [4]. The major component of Lewy bodies is α-synuclein, which is a synaptic protein with 140 amino acid residues. It has been revealed that α-synuclein plays critical roles in synaptic functions and the maintenance of synaptic plasticity. The α-synuclein fragment peptide, termed the non-amyloid component (NAC), co-accumulates with AβP in senile plaques of AD and exhibits cytotoxicity [5]. 

There are at least two possible pathogenic pathways of prion diseases; the first supports “the loss of the normal, protective functions of PrP^C^”, and the second supports “the gain of toxic functions of PrP^Sc^” [6]. Normal PrP^C^ consists of a 30–35 kDa glycoprotein anchored at the plasma membrane with a glycosylphosphatidylinositol (GPI) domain and is widely distributed throughout the body, including the liver, heart, and brain. Although the physiological roles of normal cellular PrP^C^ are not yet fully understood, knockout mice lacking PrP^C^ exhibit several neurological disfunctions, including the death of Purkinje neurons in the cerebellum, synaptic function disorder, and memory loss [7,8]. PrP^C^ reportedly regulates *N*-methyl-D-aspartate (NMDA)-type and α-amino-3-hydroxy-5-methyl-4-isoxazolepropionic acid (AMPA)-type glutamate receptors [9]. 

Increasing evidence suggests that PrP^C^ is a metal-binding protein and plays critical roles in the maintenance of metal homeostasis [10]. In the brain, various trace elements including iron (Fe), zinc (Zn), copper (Cu), and manganese (Mn) exist at different concentrations and distributions across various brain regions [11]. These trace elements, termed “neurometals”, play significant roles for brain functions as well as ubiquitous elements such as calcium (Ca) and magnesium (Mg). Recent studies supported that “neurozinc” acts as an intracellular messenger and modulates neural information [12,13]. Neurometals are essential for normal brain functions; however, their excess is neurotoxic, and therefore, their concentration and chemical form are strictly regulated. Thus, the depletion of PrP^C^ and the resulting metal dyshomeostasis may trigger neurodegenerative processes. Interestingly, PrP^C^, APP, and α-synuclein are co-localized at the synapse, which is a narrow space filled with metals and the major target of these neurodegenerative diseases. APP and α-synuclein also possess metal-binding abilities and are involved in the regulation of metal homeostasis [14].

Metals can contribute to secondary “gain of toxic function” in neurodegenerative pathways. The conformational changes and neurotoxicity of PrP^Sc^ are central for the transmission and pathogenesis of prion diseases. PrP^Sc^ as well as its peptide fragment reportedly cause synaptotoxicity and cell toxicity. Among factors that contribute to the conformational changes of proteins, metals are known to crosslink proteins by binding to several amino acids such as histidine (His), tyrosine (Tyr), arginine (Arg), and phosphorylated amino acids. We also discuss the possible mechanism of neurotoxicity induced by PrP^Sc^ and the involvements of neurometals.

Here, we review the current understanding of the link between neurometals and the two aspects of the pathogenesis of prion diseases, loss of metal-regulatory functions, and gain of toxic functions by metal-induced conformational changes, based on our studies and other new findings. 

## 2. Functions of Normal Cellular Prion Protein and Neurometals

### 2.1. Prion Protein and Copper

The link between Cu and PrP^C^ was first reported in 1997 [15]. Cu is the third most abundant metal in the brain. It plays vital roles in transmitter synthesis and myelination as a cofactor for numerous enzymes, including cytochrome C, lysyl oxidase, uricase, dopamine hydroxylase, and tyrosinase [16]. Moreover, Cu exhibits neuroprotective activity as a component of Cu/Zn superoxide dismutase (Cu/Zn SOD), which is an endogenous antioxidant. Cu is also implicated in Fe homeostasis as a component of ceruloplasmin, which is a ferroxidase. Recent studies suggest that Cu is stored in synaptic vesicles and is released into the synaptic cleft during neuronal excitation. The released Cu reportedly modulates neuronal activity by binding with NMDA-type glutamate receptors, AMPA-type glutamate receptors, and γ-aminobutyric acid (GABA) receptors [17]. Meanwhile, excess free Cu is toxic because Cu is a redox-active metal that exists as Cu^2+^ and Cu^+^ and produces reactive oxygen species (ROS). Orally digested Cu is absorbed from the gastrointestinal pathway through divalent metal transporter 1 (DMT1) and then transported by several transporters such as copper transporter 1 (CTR1) and copper-transporting ATPase (ATP7A and ATP7B). Cu deficiency or excess caused by impairments of these transporters leads to severe neurodegenerative diseases such as Wilson’s disease or Menkes disease [18]. 

Brown et al. demonstrated that PrP knockout mice exhibited decreased Cu levels in the brain and reduced activity of Cu-dependent enzymes [15]. PrP^C^ is composed of a flexible N-terminal domain and α-helix-rich C-terminal domain that is changed to a β-sheet and involved in the conformational changes of PrP^Sc^ (Figure 1). The central domain that links the N- and C-termini (residues 91–130 in humans) is conserved and critical, since the deletion of the linkage domain is reportedly lethal [19]. At the N-terminal, PrP^C^ possesses a highly conserved octarepeat domain composed of multiple tandem copies of an eight-residue sequence (PHGGGWGQ) (residues 66–90), among which His residues are critical for Cu binding. Jackson et al. reported that PrP^C^ binds to four Cu atoms in its octarepeat domain as well as two other Cu atoms in addition to two other His residues, His^96^ and His^111^ [20]. They also demonstrated that other metals including Zn^2+^, Mn^2+^, and Ni^2+^ bind to these binding sites with lower affinities compared to Cu^2+^. Walter et al. demonstrated the significance of Cu binding to His residues outside the octarepeat domain [21].

Although the role of Cu in PrP^C^ functions is still controversial, it is possible that PrP^C^ regulates Cu influx into neurons and exerts protective activity against excess Cu [22]. We have revealed that the peptide fragment corresponding to the octarepeat domain (PrP73–88) attenuates Cu-induced neurotoxicity of cultured rat hippocampal neurons [23]. Cu-bound PrP^C^ possesses SOD-like activity [24]. PrP^C^-null mice exhibit low glutathione levels and are sensitive to H_2_O_2_-induced toxicity [25]. Thus, it is possible that Cu plays essential roles in the neuroprotective functions of PrP^C^. Cu modulates the effects of PrP^C^ on the excitability of NMDA-type glutamate receptors and AMPA-type glutamate receptors [9]. Meanwhile, Cu^2+^ influences expression of the PrP^C^ gene because the gene has a metal-responsive element [26].

There is increasing evidence suggesting that Cu may be involved in the conformational conversion of PrP^C^ and the transmission of prion diseases. When Cu binds to the octarepeat region of a flexible N-terminal domain, Cu interacts and crosslinks with the α-helix-rich C-terminal domain and changes it to a β-sheet rich PrP^Sc^ [27]. Giachin et al. suggested that the non-octarepeat Cu binding site is also a key regulator of the conversion of PrP^C^ [28]. The disruption of Cu homeostasis because of a mutation of ATP7A delayed the onset of prion disease [29]. Additionally, a truncated PrP^C^ devoid of the octarepeat region exhibited lower susceptibility to PrP^Sc^ [30]. These results indicate that the regulation of Cu homeostasis is involved in the physiological roles of PrP^C^ and that Cu plays significant roles in its mechanisms of infection and neurodegeneration. 

### 2.2. Prion Protein and Zinc

Other metals including Zn, Fe, and Mn are also associated with prion diseases. Zn^2+^ has similar chemical characteristics to Cu^2+^ and shares the same binding proteins; therefore, Zn^2+^ has the next highest binding affinity to PrP^C^ compared with Cu^2+^. Zn is the second most abundant trace element in the brain and plays important roles in various physiological functions, such as mitotic cell division, immune system functioning, and synthesis of proteins and DNA, and it acts as a co-factor to more than 300 enzymes and metalloproteins [31]. In the brain, Zn is accumulated in regions such as the cerebral cortex, amygdala, hippocampus, thalamus, and olfactory cortex. Although some Zn firmly binds to metalloproteins or enzymes, a substantial fraction (approximately 10% or more) of Zn either forms free Zn ions (Zn^2+^) or is loosely bound. Chelatable Zn^2+^ is stored in the presynaptic vesicles of excitatory glutamatergic neurons and is secreted into the synaptic cleft together with glutamate during neuronal excitation. Synaptic Zn^2+^ modulates the overall brain excitability by binding to NMDA-type glutamate receptors, GABA receptors, and glycine receptors. Zn^2+^ also decreases the expression of the GluR2 subunit of AMPA-type glutamate receptors and increases calcium (Ca^2+^) and/or Zn^2+^ permeability [32]. Secreted Zn^2+^ is critical for neuronal communication, synaptic plasticity, and memory formation [33], and therefore, Zn deficiency in children results in dwarfism, delayed mental and physical development, immune dysfunction, and learning disabilities. Zn deficiency also produces learning disorders, taste disorders, and odor disorders in adults [34]. However, excess Zn^2+^ in pathological conditions such as transient global ischemia causes neuronal death and is central to the pathogenesis of vascular dementia [35]. 

Three factors are involved in the maintenance of Zn homeostasis: metallothioneins, ZnT Zn transporters, and Zrt-, Irt-like protein (ZIP) Zn transporters. Metallothioneins are ubiquitous metal-binding proteins with 68 amino acids that bind seven metal atoms (including Zn, Cu, and cadmium) via 20 cysteine residues [36]. There are three types of metallothioneins, MT-1, MT-2, and MT-3. MT-1 and MT-2 are ubiquitously expressed throughout the entire body, whereas MT-3 is primarily localized in neurons and glial cells. ZnT transporters decrease intracellular Zn via the facilitation of Zn efflux from cells [37]. There are nine types of ZnT transporters in mammals, and they are associated with the solute carrier gene family (*SLC30*). ZnT-1 is widely distributed in the brain, plays a pivotal role in Zn^2+^ efflux, and protects against excess Zn^2+^. ZnT-3 is localized to the membranes of presynaptic vesicles, transports Zn^2+^ into synaptic vesicles, and maintains high Zn^2+^ concentrations in the vesicles. ZIP transporters are another type of Zn transporter encoded by *SLC39* genes. They increase cytosolic Zn^2+^ by promoting transport from extracellular to intracellular compartments. Fourteen ZIP genes have been identified in mammals. The ZIP transporters are localized to the cell membranes or to the membranes of the Golgi apparatus or endoplasmic reticulum (ER) and control Zn^2+^ influx into subcellular organelles. ZIP8 and ZIP14 reportedly transport Fe and Mn as well as Zn [38].

Since the concentration of Zn^2+^ in the brain is much higher than that of Cu^2+^, Zn^2+^ can influence PrP^C^ binding to Cu. Spevacek et al. demonstrated that the binding of Zn^2+^ to the octarepeat domain affects the conformational changes of the C-terminal domain [39]. Thus, it is possible that Zn contributes to the conformational conversion of PrP^C^ to PrP^SC^ as well as Cu. Bioinformatics analysis has revealed the evolutionary similarities between prion genes and genes encoding ZIP transporters such as ZIP5, ZIP6, and ZIP10 [40]. Indeed, PrP^C^ reportedly co-localizes with ZIP5, as well as other ZIP proteins, and forms dimers [41]. Taylor et al. reported that ZIP6 and ZIP10 form heteromers similar to PrP structures and influence cell migration [42]. Thus, it is possible that PrP^C^ mimics ZIP transporters. Watt et al. reported that PrP^C^ acts as a Zn^2+^ sensor in the synapse and enhances the cellular uptake of Zn^2+^ via binding to the AMPA-type glutamate receptor [43]. These findings strongly suggest that PrP^C^ plays important roles in the neuronal regulation of Zn^2+^.

### 2.3. Prion Protein and Iron

Fe is the most abundant metal in the brain. Fe is essential for numerous biological functions as an enzyme cofactor for metabolic processes such as oxygen transport, oxidative phosphorylation, and energy transfer. Fe plays critical roles in brain functions such as neurotransmitter synthesis and myelination [44]. Therefore, Fe deficiency impairs learning, especially in children and infants, and it impairs working and learning ability in adults. Fe is a redox active metal and exists in two different forms, ferrous iron (Fe^2+^) and ferric iron (Fe^3+^); therefore, excess Fe can generate ROS and is toxic to neurons. 

Orally administered Fe is primarily absorbed from the gastrointestinal pathway via DMT-1 as Fe^2+^. Fe^2+^ is oxidized to Fe^3+^ by ferroxidases such as celluroplasmin, and Fe^3+^ is transported by binding to transferrin. Transferrin-bound Fe^3+^ passes through the blood–brain barrier and enters into cells via its receptors. Then, Fe^3+^ is reduced into Fe^2+^ by ferrireductase, and Fe^2+^ is transported across membranes by metal transporters and functions as a cofactor for neuronal enzymes. Thus, Fe levels as well as the Fe^2+^ to Fe^3+^ ratio are strictly regulated in normal brains.

Increasing evidence suggests that PrP^C^ is involved in Fe homeostasis. Altered Fe metabolism and reduced Fe levels in the brain were observed in PrP knockout mice [45]. Altered ferroxidase and transferrin levels in the cerebrospinal fluid (CSF) of CJD patients have also been reported [46]. PrP^C^ reportedly possesses ferrireductase activity and modulates the cellular uptake of Fe [47]. The octarepeat domain and linkage to the plasma membrane are essential for this activity. Tripath et al. demonstrated that PrP^C^ induces the conversion from Fe^3+^ to Fe^2+^, and then Fe^2+^ is intracellularly transported across membranes by the ZIP14 and DMT-1 complex [48]. 

It is widely known that the translation of various genes that possess an iron-responsive element (IRE) in their mRNA, such as ferritin or transferrin, is regulated by binding with Fe and iron regulatory proteins (IRPs) [49]. Since the mRNA of the PrP^C^ gene possesses an IRE, the Fe level controls its expression [50]. 

### 2.4. Prion Protein and Manganese

Mn is an essential trace element and crucial for various enzymes such as hydrolase, glutamine synthetase, arginase, and pyruvate carboxylase [51]. However, excess Mn is neurotoxic and induces a PD-like syndrome. Mn is absorbed by DMT-1 as well as other divalent cations and is transported by ferroportin, an iron transporter, in addition to Fe^2+^. Some ZIP transporters (ZIP8 and ZIP14) can transport Mn and Fe.

Mn is suggested to facilitate the pathogenesis of prion diseases [52]. Johnson et al. investigated the levels of trace elements in prion-infected hamster brains using X-ray photoelectron emission microscopy with synchrotron radiation and found reduced Cu and increased Mn in prion protein plaques [53]. Mn enhances the survival of PrP in model soils and increases its infectivity [54]. The risk of chronic wasting disease in elk was associated with a magnesium (Mg) deficiency and increased Mn concentrations [55]. An epidemiological survey in Slovakia suggested a relationship between the pathogenesis of CJD and the imbalance of Mn/Cu in food [56,57]. Moreover, impairment of the Mn transporter is reportedly involved in the infection process [58]. Mn influences Fe homeostasis by affecting the IRE–IRP pathway and causes accumulation of toxic Fe and increased expression of genes with IRE [59]. 

### 2.5. Other Amyloidogenic Proteins and Neurometals

Other amyloidogenic proteins such as APP and α-synuclein also possess metal-binding abilities and are involved in the regulation of metal homeostasis. APP possesses two Zn- and/or Cu-binding domains in its N-terminal and the ability to reduce oxidized Cu^2+^ to Cu^+^ [60]. Both Zn and Cu are implicated in the dimerization, trafficking, and expression of APP [61]. Cu also affects APP processing and AβP production [62]. Additionally, APP reportedly regulates Fe^2+^ efflux from cells by binding with ferroportin, which is an Fe^2+^ transporter [63]. APP mRNA contains an IRE domain similar to ferritin and other Fe-binding proteins [64]. Therefore, APP has endogenous functions in the regulation of the homeostasis of these neurometals and vice versa; thus, these neurometals can control APP expression.

α-Synuclein reportedly binds Cu^2+^, Mn^2+^, and other metal ions in its N-terminal and C-terminal domains [65]. In particular, the His^50^ residue plays a key role in the interaction between Cu and α-synuclein [66]. This His residue may play critical roles in pathogenesis because its mutation is observed in familial-type PD. Metals such as aluminum and Mn enhance the oligomerization of α-synuclein [67]. α-Synuclein possesses ferrireductase activity and converts Fe^3+^ to Fe^2+^, similar to PrP^C^, and it controls neurotransmitter synthesis by providing bioavailable Fe^2+^ to tyrosine hydroxylase and other enzymes [68]. Indeed, the Fe level and Fe^2+^ to Fe^3+^ ratio were reportedly altered in the brains of PD patients [69]. Meanwhile, α-synuclein expression is regulated by Fe levels because its mRNA possesses an IRE domain similar to APP, PrP^C^, and ferritin [70]. Mn, which has a neurotoxic profile that resembles PD, reportedly induces the overexpression of α-synuclein [71].

### 2.6. Hypothetical Scheme: Loss of Normal of PrP^C^ Function

As shown in the previous sections, all of these amyloidogenic proteins (PrP^C^, APP, and α-synuclein) possess metal-binding domains and play crucial roles in the regulation of metal homeostasis. Interestingly, they are co-localized at the synapse, where the major targets of these neurodegenerative diseases and metals are abundantly present. APP primarily exists in the presynaptic membrane and is partially present in the postsynaptic membrane [72]. PrP^C^ is located in the postsynaptic membrane with several receptors [73], and α-synuclein mainly exists in the presynaptic cytosol or membrane. ZnT-1, which facilitates the reduction of synaptic Zn levels, is also localized in the postsynaptic membrane with glutamate receptors [74].

Based on these findings, we have developed a hypothetical scheme for the interactions of these amyloidogenic proteins and neurometals at the synapse (Figure 2). During neuronal excitation, Zn^2+^ and/or Cu^2+^ are released into the synaptic clefts, and both regulate neuronal excitability by binding to glutamate receptors. The synaptic cleft is a small compartment with a radius of 120 nm and height of 20 nm, and the total volume of the synaptic clefts is estimated to be approximately 1% of the extracellular space of the brain [75]. Thus, it is plausible that Zn^2+^ and/or Cu^2+^ levels in the synaptic region may be much higher than those in the CSF. For example, in pathogenic conditions such as transient global ischemia, the Zn^2+^ concentration reportedly can reach approximately 1~100 µM [76]. 

Since an excess of Zn^2+^ is neurotoxic, ZnT-1 and PrP^C^, an analogue of the ZIP transporter, contribute to maintaining the synaptic Zn^2+^ level. Furthermore, PrP^C^ binds to Cu^2+^, regulates the intracellular Cu level, and provides synaptic Cu^2+^ to APP or other Cu-binding proteins. PrP^C^ influences AβP production by regulating the Cu^2+^ level. APP binds to Cu^2+^, reduces it to Cu^+^, and may provide Cu^+^ to CTR1, which passes Cu^+^ and functions in the intracellular accumulation of Cu [77]. Both APP and PrP^C^ contribute to the regulation of Cu levels in the presynaptic and postsynaptic regions, respectively. α-Synuclein also controls the Cu and Mn levels in the presynaptic domains. PrP^C^ acts as a ferrireductase to convert Fe^3+^ to Fe^2+^ in the postsynaptic domain and regulates Fe^2+^ influx through the DMT1 and ZIP14 complex. Meanwhile, α-synuclein acts as a ferrireductase in the presynaptic domain. Both proteins control neurotransmitter synthesis and other functions that require Fe^2+^. APP regulates Fe^2+^ efflux by binding to ferroportin. 

The distance between pre- and postsynaptic membranes (≈20 nm) is small enough for the proteins in each membrane to interact with each other. Additionally, PrP^C^ and APP are concentrated in cholesterol-rich membrane microdomains called rafts [78], which provide the platform for the interaction. PrP^C^ and AβP are reportedly co-localized in the brains of AD patients [79]. PrP^C^ functions as a receptor of toxic AβP oligomers and causes its internalization [80]. Additionally, α-synuclein reportedly influences APP processing and AβP secretion [81]. In addition to ZnT-1 and these amyloidogenic proteins, MT-3 and carnosine (β-alanyl histidine), which is synthesized and secreted from glial cells, may contribute to the maintenance of metal homeostasis at synapses [82,83]. Meanwhile, these neurometals can influence the expression of PrP and contribute the conformational conversion from PrP^C^ to PrP^Sc^ as described previously [26,27,28,39,50,59].

The crosstalk between metals and these amyloidogenic proteins is complex and delicate. When pathogenetic PrP^Sc^ enters the brain and causes a depletion of neuroprotective PrP^C^, the consequent disruption of metal homeostasis will trigger the various adverse effects observed in prion diseases. The loss of PrP^C^ will initiate oxidative damage induced by increased Cu and Fe, increase susceptibility to ROS, deplete neurotransmitters, induce synaptic and neuronal degeneration, and finally cause prion diseases. It is possible that Mn also influences crosstalk by substitution with Cu and by the accumulation of Fe, because Mn affects IRE-IRP binding [59]. It is possible that Mn enhanced neurodegeneration by inducing an overexpression of IRE-containing genes such as α-synuclein, APP, and PrP^C^. α-Synuclein is reportedly involved in Mn-induced neurodegeneration [84].

## 3. Toxic Functions of PrP and Neurometals

### 3.1. Conformational Changes and Neurotoxicity of PrP^Sc^

Regarding the second possible mechanism of “gain of toxic functions”, the conformational changes and neurotoxicity of PrP^Sc^ are central for the transmission and pathogenesis of prion diseases. PrP^Sc^ reportedly cause synaptotoxicity and cell toxicity in vitro and in vivo [85,86]. Considering the methodological difficulties of using a whole prion protein owing to its strong infectious characteristics, the peptide fragments have been widely used to investigate the neurotoxicity of PrP^Sc^. Among them, there is a peptide fragment of PrP (PrP106–126) (KTNMKHMAGAAAAGAVVGGLG); because PrP106–126 shares several characteristics with PrP^Sc^, it forms aggregates with β-sheet structures as amyloid fibrils that cause the apoptotic death of cultured neurons or glial cells, and it possesses the ability to bind to metals including Cu^2+^ and Zn^2+^ [87,88]. However, there is still controversy about the neurotoxicity of PrP^Sc^. Benilova et al. demonstrated that purified highly infectious prions are not neurotoxic, although the whole brain extracts of prion-infected mice were neurotoxic [89]. Considering that soluble oligomers of AβP are neurotoxic and AβP fibrils are nontoxic [90], the involvements of the conformation of PrP and its neurotoxicity might be complex. 

We have demonstrated that PrP106–126 forms β-sheet structures during the “aging” process (incubation at 37 °C for several days), and it exhibits enhanced neurotoxicity in primary cultured rat hippocampal neurons [23]. Since metals are the critical determinant for protein conformation, we investigated the effects of various trace elements and metal chelators on the conformational changes and neurotoxicity of PrP106–126, and we found that the co-existence of Zn^2+^ or Cu^2+^ significantly attenuated the neurotoxicity of PrP106–126. We also found that Zn^2+^ and Cu^2+^ significantly inhibited PrP106–126 oligomerization using the thioflavin T (ThT) fluorescence assay, far-ultraviolet circular dichroism (CD) spectroscopy, and atomic force microscopy (AFM) imaging. Although Cu^2+^ and Zn^2+^ reportedly facilitate the aggregation of AβP [91], our results coincide with other studies, which indicate that PrP-induced conformational changes and toxicity are attenuated by Cu [92,93]. Moreover, Cu^2+^ reportedly inhibits the aggregation of human islet amyloid peptide (amylin) [94], Thus, it is highly possible that Cu^2+^ exhibits complex effects on the oligomerization of amyloidogenic proteins. 

### 3.2. Molecular Mechanism of PrP^Sc^-Induced Neurotoxicity: Disruption of Ca Homeostasis

The apoptotic pathways induced by PrP106–126 are of great interest. PrP106–126 reportedly causes various adverse effects, such as the proliferation of microglia, induction of proinflammatory responses, ROS production, and activation of ER stress. However, the precise mechanism of neurodegeneration induced by PrP106–126 is still unclear. 

We focus here on the formation of Ca^2+^-permeable pores by the PrP106–126 peptide and the consequent Ca^2+^ dyshomeostasis. It is widely accepted that the disruption of neuronal Ca^2+^ homeostasis and alteration of the intracellular Ca^2+^ concentration ([Ca^2+^]_i_) activate various apoptotic proteins such as calpain and caspase, leading to neuronal death, and they trigger various adverse effects that are also associated with prion diseases. 

This idea was first demonstrated in a study of the neurotoxicity of AβP. In 1993, Arispe et al. first demonstrated that AβP(1–40), i.e., the first 40 residues of AβP, can directly incorporate into artificial lipid bilayer membranes and form cation-selective ion channels [95]. The channels, termed “amyloid channels”, are giant multi-level pores and can allow a large amount of Ca^2+^ to pass through them. We demonstrated the appearance of amyloid channels of AβP(1–40) on membrane patches from a neuroblastoma cell line (GT1-7 cells) as well as liposomes [96]. The activity of amyloid channels was inhibited by the addition of Zn^2+^ and was recovered by the administration of a Zn chelator, *o*-phenanthroline. In this sense, AβP might have a similar mechanism of toxicity as that underlying the toxicity of various antimicrobial or antifungal peptides that also exhibit pore-forming activity and cell toxicity. Indeed, Soscia et al. demonstrated that AβP exerts antimicrobial activity against microorganisms [97]. 

Similarly, PrP106–126 reportedly forms cation-permeable pores in artificial lipid bilayers [98]. Kourie et al. found that PrP106–126 was directly incorporated into lipid bilayers and formed cation-selective ion channels [99]. They also found that Cu^2+^ modulates the activity of PrP channels [100] and that quinacrine (a potent therapeutic drug for prion diseases) inhibited the PrP-induced currents [101]. Furthermore, other fragments of PrP in the C-terminal such as PrP82–146 and PrP90–231 also formed channels through artificial lipid bilayers [102,103]. PrP^C^ also has anti-microbial activity similar to AβP [104]. 

We observed temporal changes in [Ca^2+^]_i_ in GT1-7 cells using a high-resolution multi-site video imaging system with fura-2 as the cytosolic free fluorescent Ca reporter probe [105,106]. Shortly after exposure to PrP106–126, a marked increase in [Ca^2+^]_i_ occurred within many neurons. Another toxic fragment peptide of PrP, PrP118–135 (AGAVVGGLGGYMLGSAMS), also caused [Ca^2+^]_i_ elevation as well as oligomerization, although scrambled PrP106–126 (NGAKALMGGHGATKVMVGAAA), a nontoxic and nonamyloidogenic analogue with a random sequence of PrP106–126, did not cause such an elevation (Figure 3). We found that AβP, human amylin, and NAC also caused increases in [Ca^2+^]_i_ similar to PrP106–126. PrP106–126, AβP, and human amylin caused the perforation of liposome membranes [14,107]. Meanwhile, it is possible that PrP^C^ regulates Ca^2+^ homeostasis because PrP-null mice exhibited alterations of Ca^2+^ buffering and [Ca^2+^]_i_ [108]. Solomon et al. demonstrated that PrP lacking residues 105–125 exhibited spontaneous ion channel activity and neurodegeneration [109]. Demuro et al. reported that AβP, human amylin, PrP106–126, and polyglutamine increased [Ca^2+^]_i_ in a conformation-dependent manner [110]. Furthermore, Lashule et al. demonstrated that α-synuclein also forms annular pore-like structures similar to AβP [111].

These findings strongly suggest the “amyloid channel hypothesis”, namely that the disruption of Ca homeostasis via the upregulation of amyloid channels may be the molecular basis of the neurotoxicity of prion and other conformational diseases [14,112]. Furthermore, metals such as Zn and/or Cu are also implicated in the functions of these amyloid channels.

## 4. Conclusions

We discuss here the pathogenesis of prion diseases and focus on the interaction with neurometals based on two aspects of these diseases. PrP^C^ is a metal-binding protein and regulates the homeostasis of metals such as Cu, Zn, and Fe, as well as other amyloidogenic proteins and α-synuclein. Loss of these normal physiological functions of PrP^C^ might induce synaptic degeneration and cytotoxicity and finally cause the onset of prion diseases. Meanwhile, metals are also implicated in the conformational changes of toxic fragments of PrP peptides and the pathways of neurodegeneration. This gain of toxic function will lead to the pathogenesis of prion diseases. 

This hypothetical scheme might be beneficial in screening substances to prevent prion diseases. Clioquinol, a chelator of Zn and Cu, has been examined for possible treatment of AD [113]. Barregi et al. reported that clioquinol affected scrapie-induced memory impairment [114]. D-(−)-penicillamine, a Cu^2+^-specific chelator, reportedly attenuated the pathogenesis of prion diseases in vivo [115]. Small peptides, such as the β-sheet breaker peptide, inhibit the conformational changes of PrP and AβP [116]. Our survey of protective substances revealed that carnosine attenuated neurotoxicity induced by PrP106–126 and prevented its oligomerization [23]. Carnosine is a dipeptide composed by β-alanine and histidine that endogenously exists in muscles and brains. Carnosine has antioxidant, anti-crosslinking, and anti-glycosylation activities and the ability to bind to metals [117]. Carnosine reportedly inhibits the oligomerization of AβP, attenuates neurodegeneration in AD model mice, and inhibits Zn^2+^-induced neuronal death [118,119]. Considering these beneficial characteristics of carnosine, we have published a patent for carnosine as a possible target for drug treatment of vascular-type senile dementia [120]. Therefore, it is possible that carnosine might be a candidate for the treatment of prion diseases.

In conclusion, our results might shed light on the enigmatic roles of trace elements in the pathogenesis of prion diseases. However, further research is necessary, particularly regarding the inhibitory mechanism of carnosine and the development of possible protective agents against prion diseases.

## Figures and Tables

**Figure 1 ijms-22-01267-f001:**
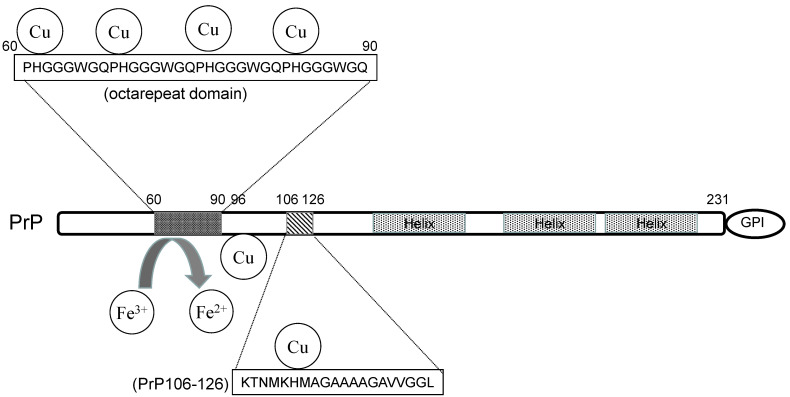
The structure and the metal-binding property of prion protein.

**Figure 2 ijms-22-01267-f002:**
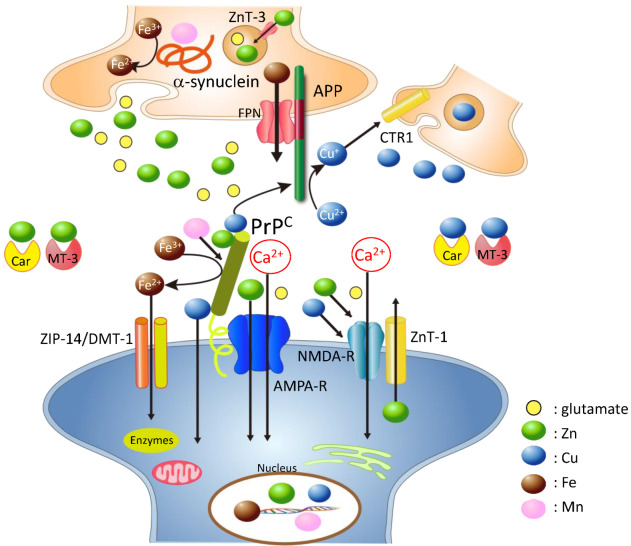
Hypothetical scheme: loss of normal functions of amyloidogenic proteins at the synapse. Normal cellular prion protein (PrP^C^) is located in the postsynaptic membrane and binds to various receptors. PrP^C^ binds to copper (Cu), zinc (Zn), and iron (Fe) and regulates their levels at the synapse. Additionally, PrP^C^ acts as a ZIP Zn transporter analogue, and the ZnT-1 Zn transporter is also localized at postsynaptic membranes; both proteins control Zn levels at the synapse. PrP^C^ can provide Cu to amyloid precursor protein (APP) or other Cu-binding proteins at the synapse. APP is mainly localized at the presynaptic membrane, binds to Cu and/or Zn, and has the ability to convert Cu^2+^ to Cu^+^. APP also regulates Fe^2+^ efflux from cells via ferroportin. α-Synuclein is mainly localized at the presynaptic domain and binds Cu, manganese (Mn), and Fe. Both PrP^C^ and α-synuclein have ferrireductase activity and provide bioavailable Fe^2+^ to enzymes at the pre- and postsynaptic regions, respectively. Fe^2+^ is transported into cells by the ZIP-14 and DMT-1 complex. Other metal-binding factors such as MT-3 and carnosine (Car) are secreted into the synaptic cleft and play critical roles in the maintenance of metal homeostasis. NMDA-R; NMDA-type glutamate receptor, AMPA-R; AMPA-type glutamate receptor, FPN: ferroportin; colored circles represent glutamate, Zn, Cu, Fe, and Mn.

**Figure 3 ijms-22-01267-f003:**
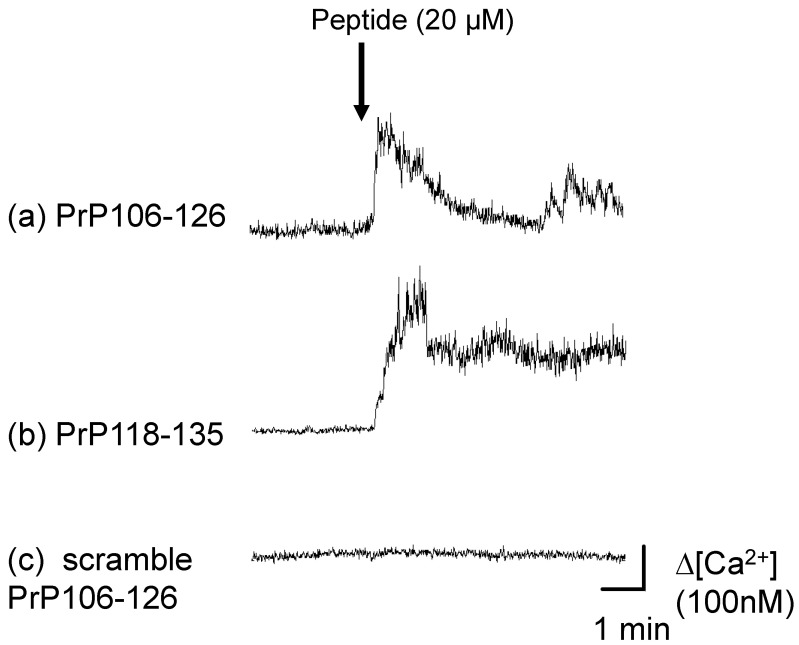
Temporal changes in intracellular calcium concentration ([Ca^2+^]_i_) in GT1-7 cells after exposure to PrP peptides. Temporal changes in fluorescence intensities corresponding to changes in [Ca^2+^]_i_ of typical GT1-7 cells before and after exposure to each PrP peptide (20 µM). (**a**) PrP106–126; (**b**) PrP118–135; (**c**) scrambled PrP106–126. The arrow indicates the time of peptide addition.

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
