# Peer review of "Neurometals in the Pathogenesis of Prion Diseases"

_ijms, 2021, doi:10.3390/ijms22031267_

Round 1

Reviewer 1 Report

In this manuscript, the authors comprehensively and concisely review the pathogenic roles of essential metals in neural cells. I think that the manuscript is worthy for publication in International Journal of Molecular Sciences, and is expected to be highly cited by future articles. I have some minor points which should be reconsidered in the revised manuscript. - The definition of “neurometals” seems to be ambiguous. The authors use the term “neurometals” for only four kinds of metal such as iron, zinc, copper, and manganese. Why does not mean calcium which is also depicted in Figure 2? - How do the authors explain the physiological role of PrP^C? It is apparent that the loss of function of PrP^C is unable to explain the pathogenic progression of prion diseases. - How do the authors explain the initial change of PrP^C to PrP^Sc? Does a metal commit to the change?

Author Response

Reply to Reviwere1

“In this manuscript, the authors comprehensively and concisely review the pathogenic roles of essential metals in neural cells. I think that the manuscript is worthy for publication in International Journal of Molecular Sciences, and is expected to be highly cited by future articles. I have some minor points which should be reconsidered in the revised manuscript.”

 Thank you very much for your kind comments and suggestions. I believe, I can polish-up the manuscript much better.

  1. “The definition of “neurometals” seems to be ambiguous. The authors use the term “neurometals” for only four kinds of metal such as iron, zinc, copper, and manganese. Why does not mean calcium which is also depicted in Figure 2? “

   Thank you very much for your comments. We mean neurometals as trace elements (exist at low concentrations in the body compared to ubiquitous element such as Ca,Mg,Na,K). They include off course more than 4 elements, but we selected and discussed here these 4 elements since the involvements of prion diseases. Thus, we add comments about neurometals in line, 106-111, add 2 references (Ref12,13). We also changed Fig.1, and changed Ca2+ to closed circle to make clear about the difference between neurometals.

  1. “How do the authors explain the physiological role of PrP^C? It is apparent that the loss of function of PrP^C is unable to explain the pathogenic progression of prion diseases. - How do the authors explain the initial change of PrP^C to PrP^Sc? Does a metal commit to the change?”

     Thank you very much for your questions. I think the mechanism of PrPC to PrPSC is not completely understood. However, I believe neurometals such as Cu and Zn may contribute some part of the processes. Cu and/or Zn reportedly binds to the octarepeat region of flexible N-terminal, and changed the structure to bind to C-terminal region, and then, causes crosslink to initiate the conformational conversion. There are several studies suggesting the Cu dyshomeostasis and the plopagation of Prion diseases. Thus, we added comments in line 168-176, and line 214-216. We also the comments in the hypothesis, line 338-339.

Reviewer 2 Report

This review paper describes the roles and known interactions of various neurometals in relation to prion diseases/prion protein and other associated protein misfolding disorders. This paper presents a balanced and up-to-date review of the known literature and its relevance to the current knowledge of this disorders.

The reviewer has small concerns about the relevance of some sections, particularly the focus on the prion peptide fragment PrP 106-126 and its physiological relevance. Recent evidence suggests misfolded prion protein is not directly neurotoxic (1). Also within the text it should be made clear e.g. within section headings that toxicity of PrP 106-126 does not equate to PrP toxicity.

References:

1. Highly infectious prions are not directly neurotoxic, Benilova et al 2020 https://doi.org/10.1073/pnas.2007406117

Author Response

Reply to Reviewer2

“This review paper describes the roles and known interactions of various neurometals in relation to prion diseases/prion protein and other associated protein misfolding disorders. This paper presents a balanced and up-to-date review of the known literature and its relevance to the current knowledge of this disorders.”

Thank you very much for your kind comments and suggestions. I believe, I can polish-up my manuscript to much better considering your suggestions.

“The reviewer has small concerns about the relevance of some sections, particularly the focus on the prion peptide fragment PrP 106-126 and its physiological relevance. Recent evidence suggests misfolded prion protein is not directly neurotoxic (1). Also within the text it should be made clear e.g. within section headings that toxicity of PrP 106-126 does not equate to PrP toxicity. References1. Highly infectious prions are not directly neurotoxic, Benilova et al 2020 https://doi.org/10.1073/pnas.2007406117”

Thank you very much for your comments, and thank you again for suggesting the recent important findings. I changed the section title 3-1 (line 350) to delete the peptide fragment, and added comments in line 353-356, and line 359-363. We also added 2 references (Ref89,90).